# SIRT7 Is a Lysine Deacylase with a Preference for Depropionylation and Demyristoylation

**DOI:** 10.3390/ijms26073153

**Published:** 2025-03-28

**Authors:** Mohammad Golam Kibria, Tatsuya Yoshizawa, Tianli Zhang, Katsuhiko Ono, Tomoya Mizumoto, Yoshifumi Sato, Tomohiro Sawa, Kazuya Yamagata

**Affiliations:** 1Department of Medical Biochemistry, Faculty of Life Sciences, Kumamoto University, Kumamoto 860-8556, Japan; kibria.bmb@gmail.com (M.G.K.); mizumoto-t@kumamoto-u.ac.jp (T.M.); ysato413@kumamoto-u.ac.jp (Y.S.); 2Cell Biology, Graduate School of Medical Science, Kyoto Prefectural University of Medicine, Kyoto 606-0823, Japan; 3Center for Integrated Control, Epidemiology and Molecular Pathophysiology of Infectious Diseases, Akita University, Akita 010-8543, Japan; zhangt@med.akita-u.ac.jp; 4Department of Microbiology, Faculty of Life Sciences, Kumamoto University, Kumamoto 860-8556, Japan; onokat@kumamoto-u.ac.jp (K.O.); sawat@kumamoto-u.ac.jp (T.S.); 5Center for Metabolic Regulation of Healthy Aging (CMHA), Faculty of Life Sciences, Kumamoto University, Kumamoto 860-8556, Japan

**Keywords:** sirtuin, SIRT7, NAD^+^, nicotinamide, deacetylation, deacylation

## Abstract

Sirtuins are nicotinamide adenine dinucleotide (NAD^+^)-dependent deacylases that remove acyl groups from lysine residues on target proteins, releasing nicotinamide. SIRT7 is associated with aging and a number of age-related diseases, but the enzymatic properties of SIRT7 are largely unknown. In the present study, we investigated the biochemical activity of SIRT7 by performing a series of in vitro kinetic studies in the presence of different acyl substrates. The binding affinity of SIRT7 for NAD^+^ was dependent on the acyl substrate, and SIRT7 showed a preference for depropionylation and demyristoylation. Nicotinamide, the end-product of the sirtuin reaction, inhibits the activity of SIRT1-6. We also found that the sensitivity of SIRT7 to nicotinamide inhibition also depended on the chain length of the acylated peptides and that nicotinamide was a poor inhibitor of SIRT7 with non-acetylated substrates. These findings may provide insights into the development of novel SIRT7 modulators for the treatment of age-related diseases.

## 1. Introduction

Lysine acetylation and acylation of proteins are post-translational modifications that regulate a wide range of biological processes [1,2]. Sirtuins (mammalian SIRT1–7) are nicotinamide adenine dinucleotide (NAD^+^)-dependent deacetylases/deacylases involved in aging and a number of age-related diseases. SIRT1–7 have a conserved NAD^+^-binding catalytic domain, whereas the sequence of the N- and C-terminal domains differs between sirtuins, partially explaining their subcellular localization and enzymatic activity [3,4]. SIRT1, SIRT6, and SIRT7 are mainly located in the nucleus, but SIRT1 and SIRT7 are also found in the cytoplasm [5,6]. In contrast, SIRT2 is predominantly cytoplasmic, whereas SIRT3, SIRT4, and SIRT5 are mitochondrial proteins [7].

SIRT1, SIRT2, and SIRT3 have strong deacetylase activity and regulate many cellular pathways, including metabolism, inflammation, and cell proliferation [4], whereas SIRT4, SIRT5, and SIRT6 have either weak or undetectable deacetylase activity in various in vitro assays. SIRT4 instead has adenosine diphosphate (ADP)-ribosyltransferase and lipoamidase activity [8,9]. A recent study showed that SIRT4 also activates methylcrotonyl-CoA carboxylase by removing dicarboxyacyl-lysine modifications and regulates insulin secretion [10]. SIRT5 has demalonylase and desuccinylase activity and plays an important role in the regulation of β-oxidation and ketone body synthesis [11,12]. SIRT6 preferentially removes long-chain fatty acyl groups, such as myristoyl and palmitoyl groups, from lysine residues and promotes tumor necrosis factor-α secretion [13,14]. SIRT7 was identified as a histone deacetylase at lysine 18 of H3 (H3K18) that regulates the transformed state of cancer cells [15], but subsequent studies have shown that SIRT7 also has multiple types of deacylase activity, including debutyrylation and desuccinylation [16]. Unlike other sirtuins, recombinant SIRT7 alone lacks efficient enzymatic activity in vitro. Intriguingly, SIRT7 can remove various acyl modifications from lysine residues when activated by interacting with DNA or RNA [17,18].

Sirtuins use NAD^+^ as a cosubstrate to remove acyl moieties from lysine residues on target proteins, releasing nicotinamide (NAM) [19]. Estimated intracellular NAD^+^ levels range from ~200 to ~500 µM depending on the cell type or tissue, and NAD^+^ levels can change by up to ~2-fold in response to various metabolic stimuli, such as glucose deprivation, fasting, calorie restriction, and exercise [19,20], suggesting that sirtuins act as metabolic sensors linking NAD^+^ availability to protein deacylation. The affinity of sirtuins for NAD^+^ is defined by the Michaelis constant, K_m_, and the K_m_ of SIRT1 for NAD^+^ is 29–94 µM [21,22]. Furthermore, the K_m_ of SIRT2, SIRT3, SIRT4, SIRT5, and SIRT6 for NAD^+^ is 18–83 µM [22,23], 98–880 µM [22,24], 35 µM [25], 980 µM [26], and 13–33 µM [27,28], respectively. These results suggest that the K_m_ for NAD^+^ is highly variable among sirtuins and that their activity may be differentially regulated by NAD^+^. Although SIRT7-DNA and SIRT7-RNA are reported to have low affinities for NAD^+^ (K_m_: 480–1270 µM) in deacetylation [29], the affinity of SIRT7 for NAD^+^ in other deacylation reactions remains to be elucidated. The activity of sirtuins is also regulated by NAM, an end-product of the sirtuin reaction. NAM inhibits sirtuins with half-maximal inhibitory concentration (IC_50_) values ranging from 30 to 200 µM, depending on the sirtuin [20,30]. However, the inhibitory effect of NAM on SIRT7 is also unknown.

SIRT7 is associated with aging and a number of age-related diseases, including obesity, type 2 diabetes, cardiovascular disease, and cancer, and it has emerged as a therapeutic target for these diseases [16,31]. Despite these advances in the identification of the biological functions of SIRT7, the lack of detailed biochemical studies has hindered the mechanistic understanding of SIRT7. In the present study, we investigated the enzymatic properties of SIRT7 by performing a series of kinetic studies. We demonstrated that the binding affinity of SIRT7 for NAD^+^ was dependent on the acyl substrate and revealed that SIRT7 showed a preference for depropionylation and demyristoylation. We also found that the sensitivity of SIRT7 to NAM inhibition also depended on the chain length of the acylated peptides and that NAM was a poor inhibitor of SIRT7 with non-acetylated substrates. These findings may provide insights that will facilitate the development of novel SIRT7 modulators.

## 2. Results

### 2.1. Catalytic Efficiency of SIRT7 on Different Acyl-Modified Peptides

SIRT1 and SIRT7 have histone deacetylase activity at lysine 9 of H3 (H3K9) and H3K18, respectively [15,32]. Mouse recombinant SIRT1 and SIRT7 were expressed in *Escherichia coli* and affinity purified (Appendix A). When the acetylated H3K9 peptide (H3K9Ac) was incubated with recombinant SIRT1, the loss of the H3K9Ac peptide and the formation of the deacetylated H3K9 peptide was detected by liquid chromatography–tandem mass spectrometry (LC-MS/MS) (Appendix A). Wild-type SIRT7, but not enzymatically inactive SIRT7^H188Y^ [33], also removed the acetyl modification from the lysine residue of histone H3K18 in the presence of RNA (Figure 1A,B). These results indicate that our in vitro assay system can adequately assess the activity of sirtuins. In addition to deacetylation, SIRT7 is reported to remove various acyl modifications, such as butyrylation, succinylation, glutarylation, octanoylation, and myristoylation, from its target proteins [16,18,34,35]. We then investigated these deacylation properties of SIRT7 in our assay system (Figure 1C–I). The same H3K18 peptide was used in all assays to interrogate only the effect of acyl groups on catalysis (Appendix A). As is consistent with previous reports, SIRT7 catalyzed the hydrolysis of butyryl, succinyl, octanoyl, and myristoyl H3K18 peptides, but hydrolysis of the glutaryl peptide by SIRT7 was not detected under our assay conditions (Figure 1D–H). In addition, SIRT7 removed propionyl and palmitoyl groups from lysine residues (Figure 1C,I). Among these acyl peptides, the hydrolysis of myristoyl and palmitoyl lysine residues was the most efficient (Figure 1J). To a lesser degree, SIRT7 hydrolyzed the butyryl group from lysine residues.

### 2.2. SIRT7 Has a Low Affinity for NAD^+^ for Deacetylation

We then performed kinetic analysis at different NAD^+^ levels in the presence of RNA, and the data were subjected to Michaelis–Menten analysis to determine the K_m, NAD_^+^ values. The K_m, NAD_^+^ value for SIRT1 with the H3K9Ac peptide was 14 µM (Figure 2A), which is close to the value (29 µM) reported previously [22]. The K_m, NAD_^+^ value for wild-type SIRT7 with the H3K18Ac peptide was 445 µM in the presence of RNA (Figure 2B), which is also consistent to the previously reported value of 480 µM [29]. Given that intracellular NAD^+^ levels range from ~200 to ~500 µM [19], this result suggests that SIRT7 activity fluctuates within the range of physiological changes in NAD^+^ levels. Damage-specific DNA-binding protein 1 (DDB1) is a target protein for SIRT7 deacetylation [36]. To investigate the biological relevance of the high K_m_ of SIRT7 for NAD^+^, HEK293T cells were transfected with a DDB1 expression plasmid alone and with a SIRT7 expression plasmid, and DDB1 acetylation was assessed by immunoprecipitation and Western blot analysis. FK866 is a highly specific inhibitor of nicotinamide phosphoribosyltransferase (NAMPT), a rate-limiting enzyme in mammalian NAD^+^ biosynthesis, and reduces intracellular NAD^+^ levels by inhibiting NAMPT [37]. In the absence of FK866, SIRT7 deacetylated DDB1 in HEK293T cells (Figure 2C, lanes 1 and 2), as previously reported. Upon stimulation with FK866, NAD^+^ levels decreased (Figure 2D) and SIRT7-mediated DDB1 deacetylation was abolished (Figure 2C, lanes 3 and 4). Consistently, the hepatic expression of *Cd36* and *Cidec*, which are upregulated by SIRT7-mediated DDB1 deacetylation [36], was significantly decreased by FK866 treatment in Hepa1–6 cells (Figure 2E and Appendix A). In contrast, the deacetylation of PPARα by SIRT1 [38], which has a low K_m_ for NAD^+^, was not abolished by the same FK866 treatment condition (Figure 2F). These results suggest that the deacetylation activity of SIRT7 is more sensitive to decreases in NAD^+^ levels than that of SIRT1.

### 2.3. SIRT7 Has a High Affinity for NAD^+^ in Depropionylation and Demyristoylation

We further investigated the kinetic parameters K_m, NAD_^+^, k_cat_ (turnover number), and k_cat_/K_m, NAD_^+^ (catalytic efficiency) for wild-type SIRT7 using different acyl-modified H3K18 peptides. The K_m, NAD_^+^ value for SIRT7 with a propionylated H3K18 peptide was lower (185 µM) than that with an acetylated peptide (445 µM) (Figure 2B and Figure 3A,G). It was reported that the K_m, NAD_^+^ value for SIRT1 tends to decrease with increasing chain length of the acyl moiety [22]. Similarly, the K_m, NAD_^+^ values for butyryl (C4), octanoyl (C8), and myristoyl (C14) peptides decreased with increasing chain length and were lowest for SIRT7 with a myristoyl peptide (153 µM) (Figure 3B,D,E,G). However, the K_m, NAD_^+^ value for SIRT7 was more variable depending on the type of acyl-modified substrate (succinyl [C4], 114 µM; palmitoyl [C16], 633 µM) (Figure 3C,F,G). Our results indicate that the affinity of SIRT7 for NAD^+^ is high in the presence of propionyl, succinyl, and myristoyl substrates.

Catalytic efficiency, k_cat_/K_m_, is considered the most physiologically relevant parameter as it reflects enzymatic activity at low substrate concentrations, mimicking conditions typically found in vivo [14,39]. As is consistent with the lower K_m, NAD_^+^ values in depropionylation and demyristoylation, the k_cat_/K_m, NAD_^+^ values for depropionylation and demyristoylation were 13 and 14 times higher than those for the acetyl peptide, respectively (Figure 3G). Although the K_m, NAD_^+^ value for SIRT7 with a succinylated substrate was also lower, the k_cat_/K_m, NAD_^+^ value was low due to its low k_cat_ value. These results suggest that SIRT7 has a preference for depropionylation and demyristoylation.

### 2.4. NAM Is a Poor Inhibitor of SIRT7 with Non-Acetylated Substrates

In addition to NAD^+^, sirtuin activity is also regulated by NAM, the end-product of the sirtuin reaction. The IC_50_ of NAM for SIRT1, 2, 3, 5, and 6 was reported to be 30–200 µM [20,40], but the IC_50_ of NAM for SIRT7 is unknown. We investigated the relative IC_50_ of NAM for SIRT7 using different acyl-modified peptides. Wild-type SIRT7 was inhibited by NAM at a low concentration (23 µM) with an acetylated peptide (Figure 4A,H), whereas its sensitivity to NAM inhibition was much lower with other acylated peptides (Figure 4B–H). In particular, NAM did not completely inhibit the desuccinylation reaction (Figure 4D). These results indicate that NAM is a potent inhibitor of SIRT7 deacetylation, but a poor inhibitor of SIRT7 with non-acetyl-type acylated substrates.

## 3. Discussion

In the present study, we established an in vitro system to assess the activity of SIRT7, and found that the K_m, NAD_^+^ value of SIRT7 for deacetylation was 445 µM in the presence of RNA, an activator of SIRT7. SIRT7 is located in both the nucleus and cytoplasm. Considering that the estimated basal NAD^+^ level is 106–118 µM in the nucleus and 74–106 µM in the cytoplasm of mammalian cells [41,42], the deacetylation activity of SIRT7 would be limited due to the insufficient availability of NAD^+^. However, SIRT7 effectively deacetylates multiple targets in vivo, and SIRT7 deficiency in mice results in resistance to obesity, fatty liver, and glucose intolerance [33,43,44]. Previous studies have shown that SIRT7 activity can be modulated by several post-translational modifications, including glycogen synthase kinase 3β-mediated phosphorylation, ataxia telangiectasia mutated and Rad3-related kinase-mediated phosphorylation, and protein arginine methyltransferase 6-mediated methylation [45,46,47]. Therefore, SIRT7 deacetylation activity may be enhanced by such intracellular modulatory mechanisms in vivo.

In addition to protein lysine acetylation, recent studies have identified an increasing number of lysine post-translational modifications by different acyl groups, and these acyl-lysine modifications have interesting biological effects [2]. To better understand the mechanisms controlling SIRT7 deacylase activity, we performed kinetic analysis and found that its K_m, NAD_^+^ values for depropionylation (185 µM) and demyristoylation (153 µM) were low, and the k_cat_/K_m, NAD_^+^ values for depropionylation and demyristoylation were 13 to 14 times higher than for deacetylation, suggesting that SIRT7 has a preference for depropionylation and demyristoylation. Tumor necrosis factor-α is modified by lysine myristoylation, and SIRT6 regulates its secretion by hydrolysis of the acylated lysine [13]. The lysine residue of Osterix, a master transcription factor for osteoblast differentiation, is propionylated, and SIRT1 stimulates its activity through depropionylation [48]. However, the proteins depropionylated and demyristoylated by SIRT7 are unknown. Further studies are needed to identify the target proteins of SIRT7 with propionyl and myristoyl modifications and the functional roles of these modifications. The reasons for the high affinity of SIRT7 for NAD^+^ in depropionylation and demyristoylation reactions are also unknown. SIRT6 has a preference for demyristoylation because it has a large hydrophobic pocket that accommodates a myristoyl group [13]. To understand the preference of SIRT7 for depropionylation and demyristoylation, the structural basis for the interaction of SIRT7 with NAD^+^ and these acyl-modified substrates would be helpful.

Sirtuins contain three spatially distinct NAD^+^-binding sites (A, B, and C sites). The adenine–ribose moiety of NAD^+^ binds to the A site, whereas the NAM moiety binds to the B site in the absence of an acylated peptide. In the presence of an acylated substrate, the binding of NAD^+^ to the B site undergoes a conformational change that moves the NAM group to the C site, where it is cleaved. The ADP-ribose product of this reaction then returns to the B site where deacylation of the acylated lysine occurs. It was proposed that free NAM binds to and blocks the internal C site, preventing the conformational change and subsequent cleavage of NAD^+^ [30,49]. We found that SIRT7 is inhibited by low concentrations (23 µM) of NAM with an acetylated peptide, which is consistent with previous findings that the IC_50_ of NAM for other sirtuins (SIRT1, 2, 3, 5, and 6) is in the range of 30–200 µM. Interestingly, the IC_50_ of NAM for SIRT7 was much higher with other acylated peptides, suggesting that the ability of NAM to inhibit SIRT7 activity is very weak in the presence of non-acetyl-type acylated substrates. The positioning of NAM at the C site of SIRT7 with non-acetylated substrates needs to be evaluated.

This study has several limitations that should be considered. First, we only investigated the enzymatic properties of SIRT7 using acyl-modified H3K18 peptides and did not evaluate them using other peptides such as DDB1, Ras-related nuclear antigen, and insulin-like growth factor 2 mRNA-binding protein [36,43,50]. Second, we found that the binding affinity of SIRT7 for NAD^+^ was high in depropionylation and demyristoylation, but the molecular mechanisms are unclear. Third, we did not investigate SIRT7-mediated depropionylation and demyristoylation in cells because the acyl-modified proteins targeted by SIRT7 are unknown. Fourth, we could not clarify the reason why NAM is a poor inhibitor of SIRT7 with non-acetylated substrates.

## 4. Materials and Methods

### 4.1. Peptides

The in vitro deacylation experiment comprised acyl peptides that corresponded to histone H3 residues 9–28 (NH_2_-KSTGGKAPRKQLATKAARKS-COOH) and 1–21 (NH_2_-ARTKQTARKSTGGKAPRKQLC-COOH). H3K18 (NH_2_-KSTGGKAPRKQLATKAARKS-COOH) and H3K18Acetyl (NH_2_-KSTGGKAPRK^Acetyl^QLATKAARKS-COOH) were synthesized by GenScript (Piscataway, NJ, USA); H3K18Propionyl (NH_2_-KSTGGKAPRK^Propionyl^QLATKAARKS-COOH), H3K18Butyryl (NH_2_-KSTGGKAPRK^Butyryl^QLATKAARKS-COOH), and H3K18Succinyl (NH_2_-KSTGGKAPRK^Succinyl^QLATKAARKS-COOH) were synthesized by Eurofins Genomics (Tokyo, Japan); H3K18Glutaryl (NH_2_-KSTGGKAPRK^Glutaryl^QLATKAARKS-COOH) was synthesized by CUSABIO Tech (Houston, TX, USA); H3K18Octanoyl (NH_2_-KSTGGKAPRK^Octanoyl^QLATKAARKS-COOH), H3K18Myristoyl (NH_2_-KSTGGKAPRK^Myristoyl^QLATKAARKS-COOH), H3K18Palmitoyl (NH_2_-KSTGGKAPRK^Palmitoyl^QLATKAARKS-COOH), H3K9 (NH_2_-ARTKQTARKSTGGKAPRKQLC-COOH), and H3K9Acetyl (NH_2_-ARTKQTARK^Acetyl^STGGKAPRKQLC-COOH) were synthesized by Toray Research Center (Tokyo, Japan).

### 4.2. Preparation of Recombinant SIRT7 and SIRT1

The purification process for mouse recombinant SIRT7 (wild-type and H188Y mutant) and SIRT1 proteins closely followed previously established protocols with some modifications [51]. Initially, the pFN18A-Halo-SIRT7 (wild-type), pFN18A-Halo-SIRT7 (H188Y), and pFN18A-Halo-SIRT1 plasmids were transformed into *E. coli* cells (K12 KRX; Promega, Madison, WI, USA). A single colony from each plasmid was cultured in LB medium with 100 µg/mL ampicillin at 37 °C until the OD_600_ reached 0.5. After further culture at 20 °C until the OD_600_ reached 0.7, induction was carried out overnight with 0.1% rhamnose (Nacalai Tesque, Kyoto, Japan). The culture was centrifuged at 6000× *g* for 10 min at 4 °C, and the pellet was resuspended in Halo purification buffer (50 mM HEPES-KOH pH 7.4, 150 mM NaCl, 1 mM PMSF, and 1% NP-40). The cells were disrupted by sonication twice for 20 s each at level 2 (Sonifier-150; Branson Ultrasonics, Danbury, CT, USA). After further incubation with 5 mM ATP and 20 mM MgCl_2_ at 37 °C for 10 min, the lysate was centrifuged at 14,000× *g* for 10 min at 4 °C. Then, 5 mL cleared lysate was incubated with 50 µL pre-washed HaloLink resin (#G1914; Promega) overnight at 4 °C. The resin was washed with Halo purification buffer, and ProTEV Plus protease (#V6101; Promega) was employed to separate the recombinant sirtuin proteins from the Halo tag.

### 4.3. In Vitro Deacylation Assay

For the SIRT7 enzymatic reaction, 8 μM acylated lysine peptide (only the glutarylated-lysine peptide was 100 µM) was incubated in 50 μL reaction buffer containing 10 mM NaCl, 10 mM KCl, 20 mM Tris-HCl pH 9.0, 1.0 mM NAD^+^ (Sigma Aldrich, St. Louis, MO, USA), 1 μg RNA, and 1 mM DTT, with or without 2.25 μg recombinant SIRT7. In the case of the SIRT1 enzymatic reaction, 8 µM peptide was incubated in 50 µL reaction buffer containing 10 mM NaCl, 10 mM KCl, 20 mM Tris-HCl pH 8.0, 1.0 mM NAD^+^, and 1 mM DTT, with or without 1 µg recombinant SIRT1. The reaction mixtures were incubated for 24 h for SIRT7 or 2 h for SIRT1 at 37 °C. The reactions were halted with 0.1% formic acid and 10% acetonitrile, followed by LC-MS/MS analysis. The deacylation efficiency of each acyl modification was determined through the conversion of acylated lysine into deacylated lysine, as previously described [52].

For enzyme kinetics with variable concentrations of the NAD^+^ substrate, an in vitro deacylation assay was performed using wild-type SIRT7 and SIRT1 with serial dilutions of the NAD^+^ substrate. The reaction was incubated at 37 °C for the appropriate time, which was within the linear part of the progression curve (60 min for SIRT7 and 10 min for SIRT1). After incubation, the reactions were quenched and subjected to LC-MS/MS analysis. Each enzymatic assay was conducted in triplicate, and the resulting data were plotted and fitted to the Michaelis–Menten equation using GraphPad Prism version 8.0.2 (GraphPad Software, Boston, MA, USA).

For the NAM inhibition assay, an in vitro deacylation assay was performed on wild-type SIRT7 with varying concentrations of NAM (Sigma Aldrich) in water (pH 7.5) at 37 °C. Following incubation, the reactions were quenched and subjected to LC-MS/MS analysis. Relative IC_50_ values were determined based on three independent assays for SIRT7 using Prism 8 (GraphPad Software) for data analysis.

### 4.4. LC-MS/MS Analysis

After reactions, the peptides were quantified by using LC-electrospray ionization-MS with an Agilent 6460 Triple Quadrupole LC-MS system (Agilent Technologies, Santa Clara, CA, USA). The samples were injected into a ZORBAX Eclipse Plus C18 column (2.1 × 50 mm, 1.8-µm; Agilent Technologies) at 45 °C, and then separated by solvent A (0.1% formic acid) and solvent B (100% acetonitrile) combination. A gradient program with solvent A and B was used: B concentration, 0 min—1%, 10 min—80%, 10.1 min—1%, 15 min—1%; and flow rate, 0.2 mL/min. The general conditions for electrospray ionization-MS were as follows: nebulizer gas, N_2_, delivered at 50 psi; nebulizer gas temperature, 250 °C; capillary voltage, 3500 V; and collision gas, G1 grade N_2_ (Taiyo Nippon Sanso Corporation, Tokyo, Japan). Appendix A provides details of the multiple reaction monitoring (MRM) parameters that were used in this study.

### 4.5. Cell Culture

HEK293T cells (Clontech Laboratories, Mountain View, CA, USA) were maintained in Dulbecco’s modified Eagle’s medium (Fujifilm Wako Pure Chemical Corp., Osaka, Japan) with 10% fetal bovine serum (Biosera, Nuaillé, France). Hepa1–6 cells (American Type Culture Collection, Manassas, VA, USA) were cultured in Dulbecco’s modified Eagle’s medium with 10% fetal bovine serum (Corning, Inc., Corning, NY, USA).

### 4.6. Plasmids

The pFN18A-Halso-SIRT7, pFN18A-Halo-SIRT7^H188Y^, pFN18A-Halo-SIRT1, pcDNA3.1-DDB1-HA, pcDNA3.1-FLAG-SIRT7, pcDNA3.1-HA-SIRT1, and pcDNA3.1-FLAG-PPARα plasmids were generated as previously described [33,51]. The expression plasmid pCl-FLAG-PCAF (plasmid #8941) [53] was purchased from Addgene (Watertown, MA, USA).

### 4.7. NAD^+^ Assay

HEK293T and Hepa1–6 cells were treated with the NAMPT inhibitor FK866 (500 pM and 8 nM, respectively), and NAD^+^ and total NAD^+^/NADH cellular levels were determined using an NAD^+^/NADH Assay Kit-WST (#N509; Dojindo Laboratories, Kumamoto, Japan). Total NAD^+^/NADH and NADH concentrations in one sample were calculated from the calibration curve, then the NAD^+^ level was determined by subtracting the NADH value from the total NAD^+^/NADH level.

### 4.8. Detection of Lysine Acetylation and Western Blotting

HEK293T cells were treated with FK866 or DMSO for 24 h and transfected with the indicated plasmids using JetPRIME reagents (Polypus, Illkirch-Graffenstaden, France) for an additional 24 h. The cells were lysed in IP buffer (20 mM Tris-HCl pH 7.4, 200 mM NaCl, 2.5 mM MgCl_2_, 0.05% NP-40, 1 mM PMSF, and protease inhibitor cocktail (Nacalai Tesque)) supplemented with 5 mM NAM and 2.5 µM trichostatin A. The lysate was incubated on ice for 20 min and subsequently sonicated twice at 4 °C with a 20 s interval at 26% amplitude using a Sonifier SFX-150 (Branson Ultrasonics) to disrupt the cells. Following sonication, the lysate was centrifuged at 14,000× *g* for 15 min at 4 °C to separate cellular debris. For immunoprecipitation, 2 mg protein from each sample was incubated with anti-HA- or anti-DYKDDDDK (FLAG)-tagged beads (Fujifilm Wako Pure Chemical Corp.) for 18 h at 4 °C with gentle rotation. After washing with IP buffer, the proteins were eluted in SDS sample buffer, and lysine acetylation was detected by Western blot with an anti-acetylated lysine antibody.

For Western blot, proteins were separated by SDS-PAGE and transferred to an Immobilon-P PVDF membrane (Merck Millipore, Burlington, MA, USA), which was probed with primary and secondary antibodies. Proteins were visualized by using Chemi-Lumi One Super (Nacalai Tesque) and a ChemiDoc Imaging System (Bio-Rad Laboratories, Hercules, CA, USA). The following primary antibodies were used: anti-DYKDDDDK (FLAG) tag (clone 1E6; Fujifilm Wako Pure Chemical Corp.), anti-HA (clone 3F10; Roche Applied Science, Penzberg, Germany), and Pan anti-acetyllysine polyclonal (TM-105; PTM Biolabs, Inc., Chicago, IL, USA).

### 4.9. Cellular Gene Expression

Hepa1–6 cells were exposed to 8 nM FK866 for 48 h, and the cells were incubated without fetal bovine serum for the final 24 h of treatment. Subsequently, total RNA was extracted from the cells utilizing the Sepasol RNA I Super Reagent (Nacalai Tesque). Quantitative reverse transcription (RT)-PCR was conducted using a PrimeScript RT Reagent Kit (RR047A; TaKaRa Bio, Inc., Kusatsu, Japan) followed by TB Green Premix Ex TaqII (RR820A; TaKaRa Bio, Inc.) and an ABI ViiA 7 Real-Time PCR System (Applied Biosystems, Inc., Foster City, CA, USA). The relative expression level of each gene was normalized to that of TATA box-binding protein (*Tbp*). The following primer sequences were used: *Cd36*, forward 5′-TTG GCC AAG CTA TTG CGA CA-3′ and reverse 5′-CTG GAG GGG TGA TGC AAA GG-3′; *Cidec*, forward 5′-GCT GAA GGG GCA GAA GTG GA-3′ and reverse 5′-GCG CTT GGC CTT GTA GCA GT-3′; and *Tbp*, forward 5′-CCC CTT GTA CCC TTC ACC AAT-3′ and reverse 5′-GAA GCT GCG GTA CAA TTC CAG-3′.

### 4.10. Statistical Analysis

The presented data are expressed as the mean ± standard deviation (SD). Statistical analysis was conducted using GraphPad Prism 8.0.2 (GraphPad Software). Statistical significance was tested using a two-tailed Student’s *t*-test for comparison between the two groups. For comparison of more than two groups, one-way analysis of variance with Dunnett’s test was used. A *p*-value less than 0.05 was considered to be statistically significant.

## 5. Conclusions

In conclusion, we have shown that the binding affinity of SIRT7 for NAD^+^ is dependent on the acyl substrate, and that SIRT7 has a preference for depropionylation and demyristoylation. We also found that the sensitivity of SIRT7 to NAM inhibition also depends on the chain length of the acylated peptides. These findings may provide insights that will help in the development of SIRT7 modulators.

## Figures and Tables

**Figure 1 ijms-26-03153-f001:**
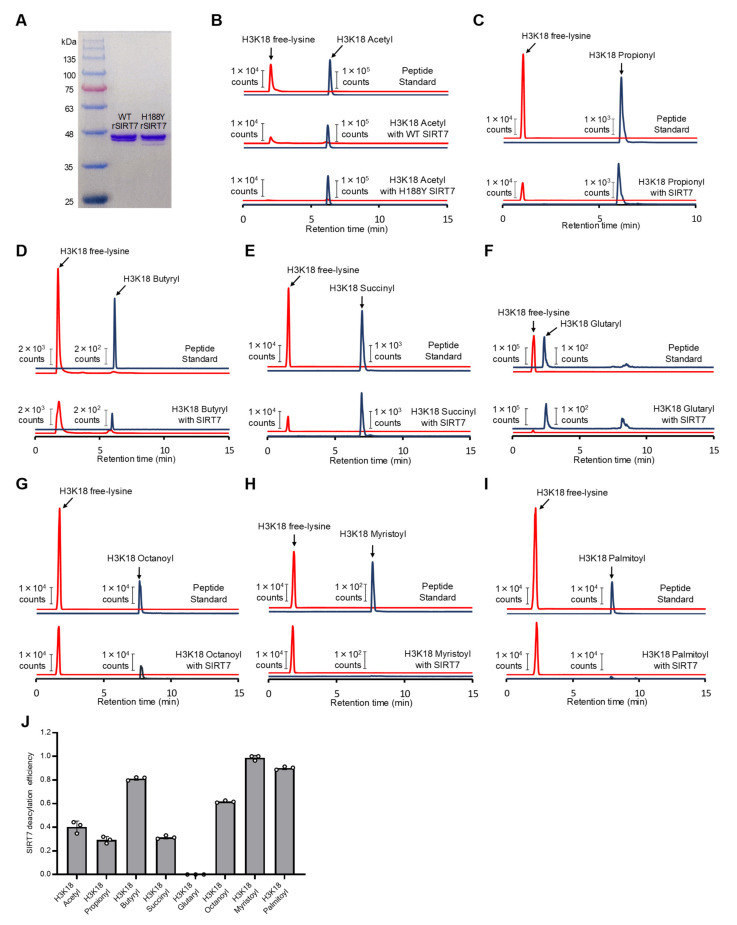
Catalytic efficiency of SIRT7 on different acyl-modified peptides. (**A**) Coomassie brilliant blue staining of recombinant SIRT7 purified from *E. coli*. (**B**–**I**) In vitro deacylation activity of SIRT7 analyzed by LC-MS/MS. Acyl group: acetyl (**B**); propionyl (**C**); butyryl (**D**); succinyl (**E**); glutaryl (**F**); octanoyl (**G**); myristoyl (**H**); and palmitoyl (**I**). (**J**) In vitro deacylation efficiency of SIRT7 for different acyl groups. Data are presented as mean ± SD of three independent experiments.

**Figure 2 ijms-26-03153-f002:**
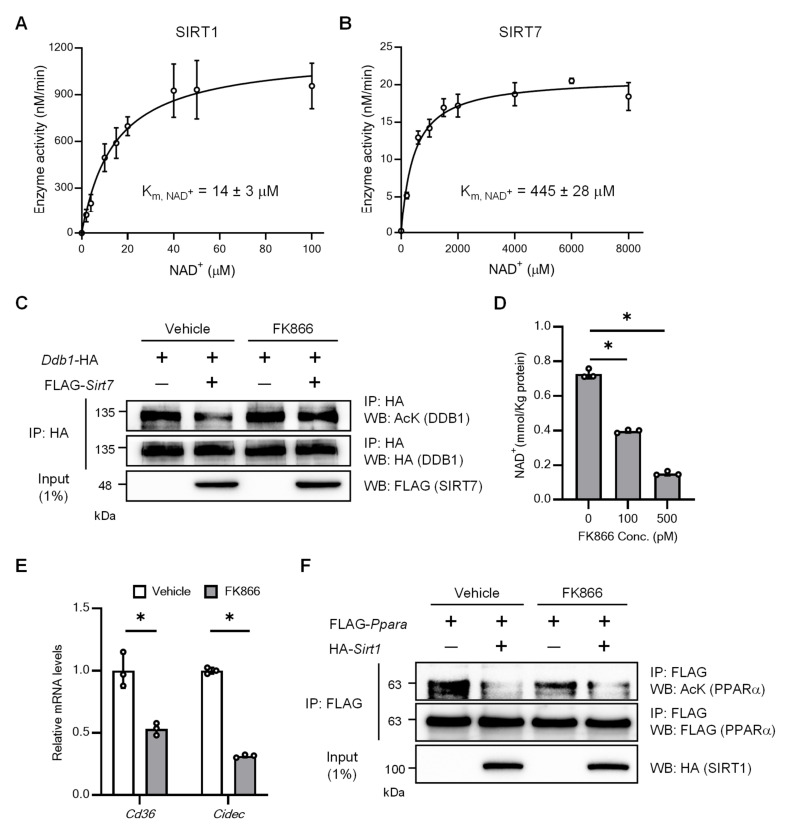
Deacetylation activity of SIRT7 is more sensitive to decreasing NAD^+^ levels than that of SIRT1. (**A**,**B**) NAD^+^-dependent SIRT1 and SIRT7 deacetylation activity. (**C**) Effect of FK866 (500 pM) on SIRT7-dependent DDB1 deacetylation. Protein lysates of HEK293T cells were subjected to immunoprecipitation (IP), after which acetylated DDB1 was detected by Western blot (WB). (**D**) Cellular NAD^+^ levels in HEK293T cells treated with indicated concentration of FK866. (**E**) Effect of FK866 (8 nM) on SIRT7-DDB1 downstream gene expression in Hepa1–6 cells. *Cd36* and *Cidec* expressions were analyzed by real-time quantitative PCR. (**F**) Effect of FK866 (500 pM) on SIRT1-dependent PPARα deacetylation. Protein lysates of HEK293T cells were subjected to IP, after which acetylated PPARα was detected by WB. Data are presented as mean ± SD of three independent experiments. One-way analysis of variance with Dunnett’s test (**D**); two-tailed Student’s *t*-test (**E**). * *p* < 0.05.

**Figure 3 ijms-26-03153-f003:**
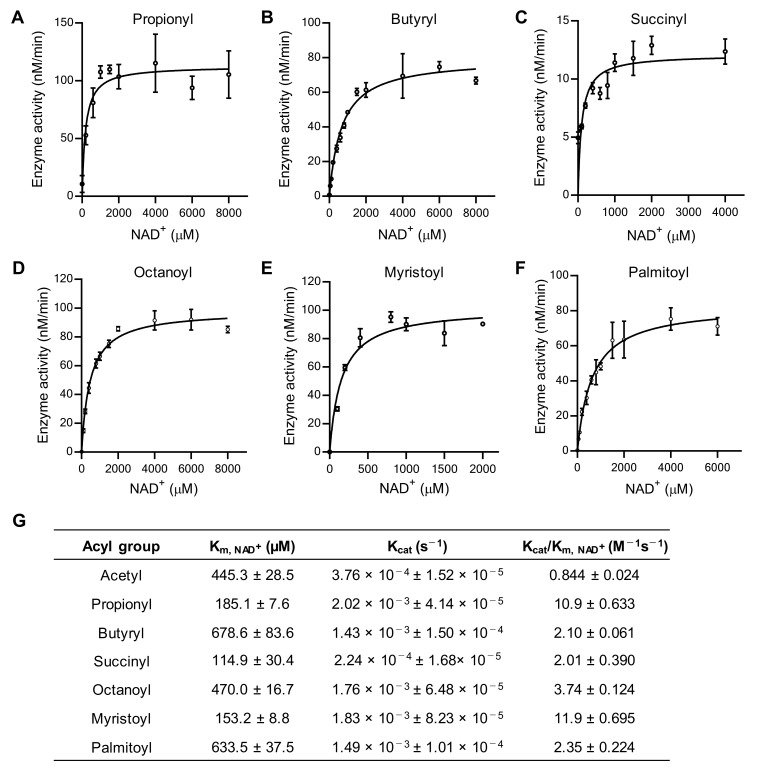
SIRT7 has a distinct affinity for NAD^+^ in different deacylation reactions. (**A**–**F**) NAD^+^-dependent SIRT7 deacylation activity. Acyl group: propionyl (**A**); butyryl (**B**); succinyl (**C**); octanoyl (**D**); myristoyl (**E**); and palmitoyl (**F**). (**G**) Kinetic parameters of NAD^+^-dependent SIRT7 deacylation. Data are presented as mean ± SD of three independent experiments.

**Figure 4 ijms-26-03153-f004:**
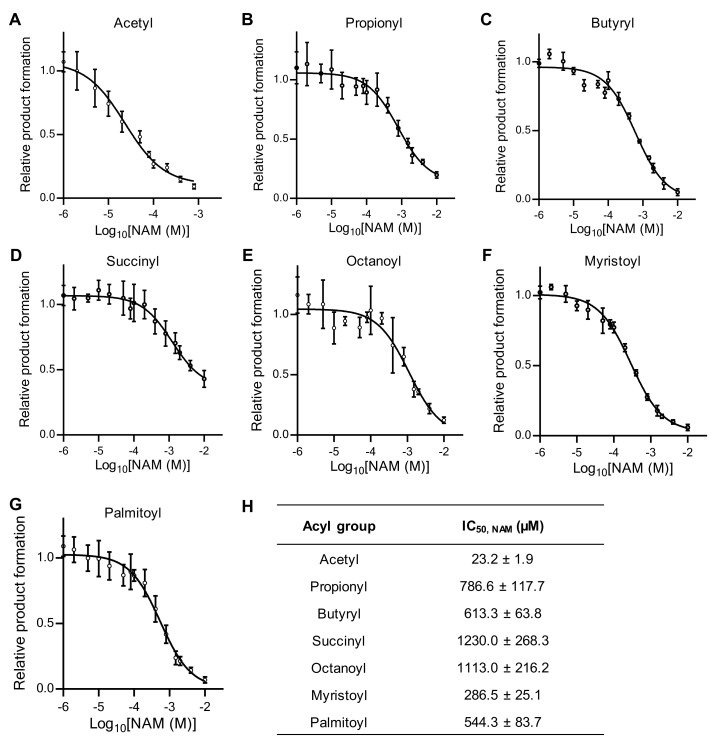
NAM is poor inhibitor of SIRT7 with non-acetylated substrates. (**A**–**G**) NAM inhibition assays showing fold change in deacylation activity. Acyl group: acetyl (**A**); propionyl (**B**); butyryl (**C**); succinyl (**D**); octanoyl (**E**); myristoyl (**F**); and palmitoyl (**G**). (**H**) Relative IC_50_ values of NAM inhibition for SIRT7 deacylation. Data are presented as mean ± SD of three independent experiments.

## Data Availability

The data presented in this study are available on request from the corresponding authors.

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
