# Peer review of "SIRT7 Is a Lysine Deacylase with a Preference for Depropionylation and Demyristoylation"

_ijms, 2025, doi:10.3390/ijms26073153_

Round 1
Reviewer 1 Report
Comments and Suggestions for Authors
The article presents the results obtained by studying the deacylation of various acylated substrates with the lysine deacylase SIRT7. The novelty is that detailed kinetic studies with SIRT7 have so far only been performed with an acetylated peptide that had a low affinity for this enzyme. The paper is well structured and written, and I recommend it for publication.
Minor comments:
- Whether the kinetic studies performed by SIRT7 wild type or H188Y mutant? This should be clarified.
- Origin of wild type should be given.
- To put the work into perspective, I suggest to upgrade the introduction of importance of the deacylation and deacetylation reaction catalysed by sirtuins
- Line 59 – „In contrast“ does not fit here
- Line 306 – delete repetition
- Lines- 308-311 – Revise the sentence since it is in contradiction.
Reviewer 2 Report
Comments and Suggestions for Authors
In this manuscript, the authors described the biochemical characterization of SIRT7 as a lysine deacylase. The authors characterized the substrate preferene, enzyme kinetics, and cellular activity of SIRT7 through comprehensive assays and formats. The outcomes from this manuscript could serve as critical references for future drug discovery and biochemistry investigation. Therefore, the manuscript is recommended for publication with the following minor revisions:
- In Figure 1F, the free H3K18 peak seems comparable or higher than the inactive SIRT7 control in Figure 1B, but the catalytic efficiency was quantified as 0 in Figure 1J. Can the authors double check the quantification?
- A small bump was observed in Figure 1H but not in Figure 1E, G, and I. Assuming the authors used the same standard peptide, is this a side product from SIRT7 or something else? Can the authors clarify?
- In Figure 2F, FK866 treatment seems to lower the endogenous Ac-PPARa level without SIRT1. Does the cell also transfected with SIRT7? The decrease in the endogenous level may impact the conclusion that FK866 did not affect PPARa deacylation.
- In Figure 4D, it seems that NAM cannot fully inhibit de-succinylation reaction as the lower limit seems to approach 0.2 or 0.3. Although this does not impact the overall conclusion of the study, it is recommended that the authors fit the curve to "relative IC50" or mention this "abnormality" in the main text.
